# Epidemiology and Molecular Characterization of Zoonotic Gastrointestinal Protozoal Infection in Zoo Animals in China

**DOI:** 10.3390/ani14060853

**Published:** 2024-03-10

**Authors:** Diya An, Tingting Jiang, Changsheng Zhang, Lei Ma, Ting Jia, Yanqun Pei, Zifu Zhu, Qun Liu, Jing Liu

**Affiliations:** 1National Animal Protozoa Laboratory, College of Veterinary Medicine, China Agricultural University, Beijing 100193, China; 18848892716@163.com (D.A.); jtt_aau@163.com (T.J.); 18349325943@163.com (Y.P.); s20193050765@cau.edu.cn (Z.Z.); qunliu@cau.edu.cn (Q.L.); 2National Natural History Museum of China, Beijing 100050, China; zhangdoudou77@sina.com; 3College of Life Science, Hebei Normal University, Shijiazhuang 050024, China; lmahappy@hebtu.edu.cn; 4Beijing Key Laboratory of Captive Wildlife Technologies, Beijing Zoo, Beijing 100044, China; jiating_2005@163.com; 5Key Laboratory of Animal Epidemiology of the Ministry of Agriculture and Rural Affairs, College of Veterinary Medicine, China Agricultural University, Beijing 100193, China

**Keywords:** gastrointestinal protozoan, zoo animals, genotype, zoonoses

## Abstract

**Simple Summary:**

Zoo visitors frequently interact with animals, heightening the potential for the transmission of zoonotic parasitic diseases between humans and animals. This study aimed to assess the prevalence of zoonotic gastrointestinal protozoa in animals from five cities in China, elucidating the species and infection rates of these parasites. The findings revealed a high incidence of intestinal parasitic protozoal infections in zoo animals, with the identified zoonotic species and genotypes including *Cryptosporidium* spp., *Giardia duodenalis*, *Enterocytozoon bieneusi*, and *Blastocystis* spp. The imperative to prevent and control parasitic diseases in zoos extends beyond the realm of protection and management; it holds significant public health implications.

**Abstract:**

Zoo animals, harboring zoonotic gastrointestinal protozoal diseases, pose potential hazards to the safety of visitors and animal keepers. This study involved the collection and examination of 400 fresh fecal samples from 68 animal species, obtained from five zoos. The aim of this study was to determine the occurrence, genetic characteristics, and zoonotic potential of common gastrointestinal protists. PCR or nested PCR analysis was conducted on these samples to detect four specific parasites: *Cryptosporidium* spp., *Giardia duodenalis*, *Enterocytozoon bieneusi*, and *Blastocystis* spp. The overall prevalence of *Cryptosporidium spp* was 0.5% (2/400), *G. duodenalis* was 6.0% (24/400), *Blastocystis* spp. was 24.5% (98/400), and *E. bieneusi* was 13.5% (54/400). *G. duodenalis*, *Blastocystis* spp., and *E. bieneusi* were detected in all of the zoos, exhibiting various zoonotic genotypes or subtypes. *G. duodenalis*-positive samples exhibited three assemblages (D, E, and B). *Blastocystis* spp. subtypes (ST1, ST2, ST3, ST4, ST5, ST8, ST10, ST13, and ST14) and one unknown subtype (ST) were identified. A total of 12 genotypes of *E. bieneusi* were identified, including SC02, BEB6, Type IV, pigEBITS 7, Peru8, PtEb IX, D, CD9, EbpC, SCBB1, CM4, and CM7. Moreover, significant differences in the positive rates among different zoos were observed (*p* < 0.01). The findings indicate that zoo animals in China are affected by a range of intestinal protozoa infections. Emphasizing molecular identification for specific parasite species or genotypes is crucial for a better understanding of the zoonotic risk. Preventing and controlling parasitic diseases in zoos is not only vital for zoo protection and management but also holds significant public health implications.

## 1. Introduction

The confined living environment of captive animals in zoos, distinct from their natural habits, renders them more susceptible to diseases due to limited activity space and high population density [1]. Pathogen infections in those animals not only adversely affect their health and well-being but also pose potential threats to zoo staff. Furthermore, the escalating number of visitors to wildlife parks heightens the risk of direct or indirect contact between tourists and wild animals. This contributes significantly to the widespread occurrence of zoonotic gastrointestinal parasite infections among zoo animals. Therefore, a comprehensive understanding of the health and welfare of wildlife in zoos is crucial. This understanding serves not only the purpose of conservation and management but also is imperative for safeguarding public health [2].

As a consequence of regular deworming and the implementation of hygienic measures, helminth infections are infrequent. However, certain protozoan parasites, such as *Cryptosporidium* spp., *G. duodenalis*, *E. bieneusi*, and *Blastocystis* spp., are commonly reported and recognized as significant contributors to gastro-enteritis [3]. These common gastrointestinal protozoa have a global distribution and have the potential to infect various hosts, including humans, livestock, companion animals, and wildlife, primarily through the fecal–oral route. Additionally, transmission can occur through the ingestion of contaminated food and water [4]. Symptoms in immunocompetent hosts are generally mild and self-limiting, whereas immunocompromised or deficient hosts may experience severe chronic diarrhea, malnutrition, and even face the risk of death [5].

Several studies have consistently reported the prevalence of *Cryptosporidium* spp. and *Giardia* spp. in eastern Europe. *Cryptosporidium* spp. and *G. duodenalis* have been ranked as the sixth and eleventh most important foodborne parasites globally, respectively [6]. Although fatalities caused by *G. duodenalis* are uncommon, fatal cases have been reported in chinchillas and birds [7]. Additionally, multiple investigations into *Cryptosporidium* spp. have demonstrated that *C. andersoni* is the fourth major species infecting humans, alongside the commonly known *C. hominis*, *C. parvum*, and *C. meleagridis* species [8]. One study described a healthy man who was found to be infected with a new microsporidium species [9]. While the source and mode of 

Transmission for this microsporidium infection remain uncertain, it is hypothesized that humans or animals infected with microsporidium may be the likely sources of infection. At the same time, another report identified *E. bieneusi* in 11 adults (13.25%) and 23 children (13.61%) suffering from diarrhea [10]. *Blastocystis* sp., a controversial unicellular protist, has been documented to inhabit the gastrointestinal tract of humans and various animal species around the world [11]. Likewise, a study conducted in a rural community in Nepal revealed the presence of *Blastocystis* sp. subtype 4 in humans, the animals they raised, and the rivers they regularly frequented [12].

Polymerase chain reaction (PCR) is the most modern practical technology in diagnosis, and compared with classical techniques, it has been shown to be more rapid, with results obtained in a few hours, and also more reliable [13]. In addition, PCR can overcome the interference caused by the persistent presence of parasite antigens and antibodies, which can be applied to clinical diagnosis and epidemiological investigation. At the same time, PCR can also be used for genotyping based on a stable marker, DNA, and is not dependent on gene expression. The ability to distinguish between genomes is important in several disciplines of microbiological research, for example, in studies on population genetics and microbial epidemiology [14].

Considering that humans and animals are in constant interaction with their environment, epidemiological studies of those four zoonotic protozoans are indispensable. However, the genetic diversity and potential for the zoonotic transmission of these species of intestinal parasitic protozoa among captive wildlife remain largely unexplored. Therefore, the primary objective of this study was to evaluate the prevalence of zoonotic intestinal protozoa in zoo animals and to investigate the presence of specific species and subtypes within zoos in China.

## 2. Materials and Methods

### 2.1. Specimen Collection

A total of 400 fecal samples were randomly collected from 68 species of animals in Guiyang city (*n* = 49), Beijing city (*n* = 101), Shijiazhuang city (*n* = 69), Tangshan city (*n* = 66), and Xingtai city (*n* = 115) in China from September 2020 to November 2021. The animals were classified into eight categories, including 84 fecal samples from 14 primate species, 171 fecal samples from 22 artiodactyla species, 37 fecal samples from 5 perissodactyla species, 6 fecal samples from 1 proboscidean species, 5 fecal samples from 1 marsupial species, 2 fecal samples from 2 avian species, 89 fecal samples from 21 carnivora species, and 6 fecal samples from 2 rodent species. Upon collection, the samples were immediately placed in dry, clean, and labeled self-sealing bags. Subsequently, they were transported to the National Protozoa Laboratory, China Agricultural University, and stored at 4 °C until laboratory examination.

### 2.2. Genomic DNA Extraction

Genomic DNA was extracted from approximately 200 mg of fecal samples using the TIANamp Stool DNA Kit (Tigen, Beijing, China, TIANamp Stool DNA Kit) according to the manufacturer’s instructions. The elution buffer (50 μL) was used to elute the DNA, which was subsequently stored at −20 °C for PCR amplification.

### 2.3. PCR Amplification

The 400 samples were identified by nested PCR for *Cryptosporidium* spp., *G. duodenalis*, and *E. bieneusi*, as well as common PCR for *Blastocystis* spp. The small subunit ribosomal RNA gene (*SSU rRNA*) was amplified to identify *Cryptosporidium* spp. and *Blastocystis* spp. For *E. bieneusi* identification, the internal transcribed spacer (*ITS*) sequence was amplified, while the triose-phosphate isomerase (*tpi*) and β-giardin (*bg*) gene were amplified for *G. duodenalis* identification. Primer sequences for the four intestinal protozoa species are shown in Table 1.

The first-round template of nested PCR was the extracted DNA, with a reaction system volume of 20 μL. The second-round template consisted of the product from the first round of PCR, diluted 10 times, with a reaction system volume of 25 μL. The amplified PCR products were analyzed using 1.5% agarose gel electrophoresis and visualized by staining with Golden View. A gel imaging analysis system and UV light were used to observe the electrophoresis results.

### 2.4. Sequencing and Phylogenetic Analysis

The positive PCR products were sent to Beijing Ruiboxingke Company for sequence analysis. The obtained sequences were then compared with published GenBank sequences using the freely available Basic Local Alignment Search Tool (BLAST) provided by the National Center for Biotechnology Information (NCBI) (https://blast.ncbi.nlm.nih.gov/Blast.cgi, accessed on 2 June 2021). The sequence alignment with the downloaded reference sequences was analyzed using Clustal X 2.13 software to determine the species/genotype of intestinal protozoa.

The *ITS* sequence obtained in this study, along with a reference sequence of *E. bieneusi*, was used to construct the phylogenetic tree using Mega6.0 software. The neighbor-joining (NJ) method and the Tamura–Nei model was selected as the appropriate model to analyze the phylogenetic relationship. To assess the reliability analysis of the evolutionary tree, bootstrap analysis with 1000 replicates was performed. The reference sequences necessary for constructing the evolutionary tree were obtained from the previously reported literature and downloaded from GenBank.

### 2.5. Statistical Analysis

The chi-square test was used to calculate the differences using SPSS 20.0 software. A statistical significance level of *p* < 0.01 was considered to indicate a significant difference.

## 3. Results

### 3.1. Prevalence and Species Distribution of Cryptosporidium spp.

The prevalence and species distribution of *Cryptosporidium* spp. were investigated in a sample of 400 zoo animals collected from five different zoos in China. Among the collected samples, two individuals tested positive for *Cryptosporidium* spp., resulting in an overall positive rate of 0.5% (2/400). The positive samples originated from a camel in Guiyang and an argali in Beijing. Subsequent analysis identified the *Cryptosporidium* spp. present in these positive samples as *C. andersoni* and *C. ubiquitum*.

### 3.2. Prevalence and Genotype Distribution of G. duodenalis

Among the 400 samples included in this study, PCR examination detected the presence of *G. duodenalis* in 6.0% of the samples (24/400). The highest positive rate was observed in Guiyang Wildlife Park at 16.3% (8/49), while the lowest positive rate was observed in Xingtai Zoo at 0.9% (1/115). The other three zoos exhibited positive rates of 9.9% (10/101) in Beijing, 4.5% (3/66) in Tangshan, and 2.9% (2/69) in Shijiazhuang. The study found statistically significant differences in the positive rates of *G. duodenalis* among the various zoos (*p* < 0.01). Table 2 presents the distribution of *G. duodenalis* detection across different zoos. Out of the 24 samples that tested positive for *G. duodenalis*, three different genotypes were identified: assemblage D, E, and B. Based on the *tpi* gene, a total of 14 positive samples of *G. duodenalis* were identified, with assemblage E (*n* = 2) and B (*n* = 12) being the predominant genotypes. Based on the *bg* gene, 17 positive samples of *G. duodenalis* were found, including assemblage E (*n* = 12), B (*n* = 4), and D (*n* = 1). According to Table 3, the prevalence of single infections (91.7%, 22/24) was higher than that of mixed infections (8.3%, 2/24) among zoo animals. Among the 24 positive samples, 11 samples were infected with assemblage E, including giraffes (*n* = 2), a milk goat (*n* = 1), Siberian ibex (*n* = 2), an argali (*n* = 1), roe deer (*n* = 2), an addax (*n* = 1), a yak (*n* = 1), and a ring-tailed lemur (*n* = 1). Furthermore, 10 samples demonstrated single infections with assemblage B, which consisted of ring-tailed lemurs (*n* = 8) and chimpanzees (*n* = 2). Only one sample from a masked civet (*n* = 1) exhibited a single infection with assemblage D. Additionally, mixed infections with both assemblage E and B were observed in only two samples from a milk goat (*n* = 1) and brown bear (*n* = 1) (*p* < 0.05). In addition, the highest prevalence of *G. duodenalis* infection was found in primates (36.7%), while the lowest prevalence was found in carnivores (22.2%). These findings suggest the importance of prioritizing prevention and control measures for *G. duodenalis* within primate habitats.

### 3.3. Prevalence and Gene Subtype Distribution of Blastocystis spp.

A total of 98 samples tested positive for the presence of *Blastocystis* spp. through PCR analysis, resulting in an overall positive rate of 24.5%. Among the zoos, Beijing had the highest prevalence of *Blastocystis* spp. with a rate of 35.6% (36/101), while Tangshan Zoo had the lowest prevalence at 10.6% (7/66). The other three zoos had the following prevalence rates: Guiyang with 24.5% (12/49), Shijiazhuang with 20.3% (14/69), and Xingtai with 25.2% (29/115). There was a significant difference in the prevalence of *Blastocystis* spp. infections among the different zoos (*p* < 0.01).

*Blastocystis* spp. was detected in a total of 68 animals from six different animal groups: proboscidean, artiodactyla, perissodactyla, primate, carnivora, and avian species. However, no detection of *Blastocystis* spp. was found in rodents or marsupials. Among the animal groups, the infection rates for *Blastocystis* spp. were as follows: artiodactyla, 35.7% (61/171); perissodactyla, 5.4% (2/37); avian, 50.0% (1/2); carnivore, 3.4% (3/89); primate, 35.7% (30/84); and proboscidean, 16.7% (1/6). There were significant differences in the prevalence of *Blastocystis* spp. infections among the six animal groups. Based on the data provided, the data suggest that enhancing control measures for *Blastocystis* spp. is imperative in the feeding and management practices of artiodactyla. Table 4 provides an overview of the detection of *Blastocystis* spp. in various animals.

By conducting sequence alignment, a total of nine known *Blastocystis* spp. subtypes (ST1, ST2, ST3, ST4, ST5, ST8, ST10, ST13, and ST14) and one unknown subtype (ST) were identified by sequence alignment. Among these subtypes, the most prevalent was ST10, found in 31 out of 98 positive samples. ST10 was detected in 14 different species, making it the dominant subtype. Table 5 provides the distribution of the different *Blastocystis* spp. subtypes among the animals.

### 3.4. Prevalence and Genotype Distribution of E. bieneusi 

The *ITS* sequence of *E. bieneusi* was detected through PCR analysis, and out of the 400 samples tested, 54 were positive, resulting in an overall infection rate of 13.5% (54/400). Xingtai Zoo had the highest positive rate of *E. bieneusi* at 25.4% (29/115), while Tangshan Zoo had the lowest positive rate at 6.1% (4/66). The infection rates in Guiyang, Beijing, and Shijiazhuang were 12.2% (6/49), 6.9% (7/101), and 11.6% (8/69), respectively. Among these, the infection rate of carnivores was 17.8%, and the infection rate of artiodactyls was 15.8%, ranking first and second, respectively. Hence, attention should be directed towards controlling *E. bieneusi* in the residential areas of these two animal groups.

The study found a significant difference in the positive rate of *E. bieneusi* among the different zoos (*p* < 0.01). A total of 12 genotypes were identified in the 54 *E. bieneusi*-positive samples. These genotypes included SC02 (*n* = 1) found in a panda; BEB6 (*n* = 8) found in milk goats, giraffes, addax and sika deer; Type IV (*n* = 7) found in Suri alpacas (*Lama glama)*, roe deer, gibbons, ring-tailed lemurs, bush pigs, alpacas (*Vicugna pacos*), and red deer; pigEBITS 7 (*n* = 2) found in a gibbon and Siberian ibex; Peru8 (*n* = 2) found in a river deer and masked civet; PtEb IX (*n* = 3) found in gray wolves; D (*n* = 11) found in colobus monkeys (*Colobus polykomos*), squirrels, African lions (*Panthera leo*), masked civets, camels, yaks, fallow deer, sika deer, red deer, and gray wolves; D9 (*n* = 15) found in yaks, fallow deer, sika deer, big-eared sheep, red deer, wild horses, fragrance pigs, Amur tigers (*Panthera tigris ssp. altaica)*, and gray wolves; EbpC (*n* = 1) found in a brown bear; SCBB1 (*n* = 2) found in black bears; CM4 (*n* = 1) found in a squirrel; CM7 (*n* = 1) found in an argali.

The obtained sequences of the 12 *E. bieneusi* genotypes were utilized to construct neighbor-joining phylogenetic trees (Figure 1). The phylogenetic analysis revealed the classification of the genotypes into different groups based on their genetic relatedness. Genotype D, PigEBITS7, Peru8, Type IV, SC02, and EbpC were classified into group 1, which suggests potential for zoonotic transmission. CM7, BEB6, and SCBB1 were categorized into group 2, indicating a potentially zoonotic nature. CM4 was assigned to group 9, while CD9 and PtEb IX were placed in group 11 based on the phylogenetic analysis.

## 4. Discussion

In the current study conducted on Chinese zoo animals, the prevalence rates of *Cryptosporidium* spp., *G. duodenalis*, *E. bieneusi*, and *Blastocystis* spp. were observed. The overall infection rates are presented in Table 6, along with the identification of various zoonotic genotypes. It is strongly advised that both breeders and visitors exercise caution during interactions with these animals, particularly when there is a possibility of contact with their feces and wastewater. This precaution is crucial for preventing the transmission of zoonotic parasitic diseases.

Advances in molecular biology detection technology have provided significant advantages in comprehending the classification and population genetic characteristics of various intestinal protozoa. *Cryptosporidium* spp., which exhibit a broad host range, have been reported in animals from various zoos in China [19]. In this study, two samples tested positive for *Cryptosporidium* spp. were identified, including *C. andersoni* found in a camel and *C. ubiquitum* found in an argali. *C. andersoni* was previously considered to primarily infect domestic animals with certain host specificity, but it has been identified in various animal hosts as well as humans [20]. Similarly, *C. ubiquitum* has been identified in argali and even in drinking water, suggesting it serves as a potential source of *Cryptosporidium spp* infection in humans [21].

Currently, six species of *Giardia* have been identified, including *G. duodenalis*, *G. agilis*, *G. ardeae*, *G. muris*, *G. microti*, and *G. psittaci*. Among these, *G. duodenalis* is the most commonly reported species in humans and animals. *G. duodenalis* is considered as a complex species and has been further categorized into eight assemblages (A–H) based on genetic analysis [22]. Assemblages A and B have a wide host range and are responsible for human infections [23]. The remaining six assemblages usually occur in specific animal hosts, although assemblages C, D, E, and F have been occasionally detected in human cases as well [24]. In total, 27 parasite species and over 70 host-adapted genotypes have been identified [25]. By PCR amplification of the *tpi* and *bg* genes of *G. duodenalis*, positive results were observed in 11 species from three animal groups. Three assemblages, E, B, and D, were identified among these positive samples. The detection of assemblage B in ring-tailed lemurs and chimpanzees confirms previous reports of its prevalence in primates [26]. Additionally, a mixed infection of assemblage E and B was found in milk goats and brown bears. Utilizing multiple genetic loci for the detection of *G. duodenalis* can improve the sensitivity of the diagnosis of mixed infections, with clusters of *G. duodenalis* being more commonly seen in some developing countries [27]. The identification and characterization of different *Giardia* species and assemblages are crucial for understanding their epidemiology, host range, and potential for zoonotic transmission. By studying the genetic diversity of *Giardia*, it becomes possible to unravel the complex interactions between the parasite, animals, and humans and design appropriate control measures to prevent the spread of infection.

*E. bieneusi* is classified as a microsporidium species within the fungi group, but it is often studied alongside other zoonotic protozoa. *E. bieneusi* is the most frequently diagnosed microsporidium species and accounts for more than 90% of human microsporidiosis cases [28]. While *E. bieneusi* infection in wild animals has been extensively reported worldwide, there have been relatively few studies on its occurrence in zoo animals [29]. A total of 12 genotypes of *E. bieneusi* were identified in this study. Previous reports have shown that genotypes within group 1 (D, PigEBITS7, Peru8, Type IV, SC02, and EbpC) have been detected in humans and various animal groups, indicating low host specificity and high potential for cross-species transmission [30,31]. Additionally, emerging evidence indicates that genotypes within group 2 also possess zoonotic potential. Genotypes I, J, BEB4, and BEB6 from group 2 have been found to infect humans [32]. The four genotypes, SC02, SCBB1, CM4, and PtEb IX, that we detected were found to be the dominant genotypes among pandas, black bears, squirrels, and gray wolves, respectively. The zoonotic genotypes D, PigEBITS7, Peru8, Type IV, SC02, and EbpC were identified in a variety of animals [33,34,35,36], suggesting that zoo animals may be important sources of human *E. bieneusi* infections and should be closely monitored and controlled.

Out of the 400 samples tested, a total of 98 (24.5%) were found to be positive for *Blastocystis spp*. The prevalence of *Blastocystis* spp. varied across different animal species: it was 35.7% in artiodactyla and primate species, 50% in avian species, 16.7% in proboscidean species, 5.4% in perissodactyla species, and 3.4% in carnivora species. No *Blastocystis* spp. were detected in rodents or marsupials. Another study conducted in two zoos in France reported a total positive rate of 32.2% for *Blastocystis* spp. [36], while a higher infection rate of 40.2% was reported in 295 wild animals in Qinling Mountains, China [37]. Recent molecular epidemiological studies have revealed potential modes of transmission for *Blastocystis* spp., including human-to-human, foodborne, waterborne, and zoonotic routes. Notably, the World Health Organization classified *Blastocystis* sp. as a waterborne pathogen, implying a potential public health concern [11]. *Blastocystis* spp. comprise 17 subtypes (ST1-ST17), with ST1-ST9 capable of infecting both humans and a variety of animals [38]. In China, *Blastocystis* spp. infection is primarily caused by ST1-ST3 subtypes in humans, while ST10-ST17 subtypes predominantly affect animals. This infection has been reported in more than 12 provinces/municipalities in humans and over 25 different animal hosts, showing significant variability in prevalence across geographic regions [10]. Nine *Blastocystis* sp. subtypes, including ST1-ST5, ST8, ST10, ST13, ST14, and one unidentified subtype (Unknown ST), were identified across 14 different animal species. Among those subtypes, ST1-ST5 and ST8 have previously been reported to infect humans. Specifically, ST1-ST4 and ST8 are predominantly found in primates, while ST5 is primarily detected in ungulates, which is consistent with previous studies [39,40]. ST3 merged as the dominant subtype in human infections, and ST1 also exhibited significant presence among human infections.

In general, the prevalence of *Blastocystis* spp. infections was found to be the highest, while *Cryptosporidium* spp. had the lowest prevalence. The variation in infection rates of those protozoans among studies can be attributed to several factors, including geographical location, climatic conditions, sampling season, animal species, sample size, and most importantly, the zoo policies related to feeding and hygiene management. For example, our study highlights the unique living conditions at Guiyang Wild Animal Park, which employs a semi-free-range breeding mode, unlike the other four zoos. This arrangement fosters increased interaction among animals. Our findings reveal that the *Cryptosporidium* spp. and *G. duodenalis* infection rates at Guiyang Wild Animal Park are notably higher compared to the other locations. Similarly, the prevalence of *E. bieneusi* and *Blastocystis* spp. infection is also considerable, ranking second and third, respectively. These results suggest a potential correlation between zoo living patterns and parasite infection rates. The presence of these parasites can be attributed to their relatively simple lifecycle, which does not involve intermediate hosts and allows for immediate infectivity upon excretion. Additionally, their low infective dose and short prepatent period contribute to their ease of transmission.

Our data showed the presence and diversity of significant opportunistic protozoa in zoo animals. Furthermore, we employed both the saturated saline floating method and centrifugal precipitation method to enhance the detection of oocysts for microscopic examination and identified the presence of nematodes oocysts.

However, the origin, the route of the transmission, and the significance of these pathogens for both animal and public health remain unknown. Addressing these lingering questions requires a more comprehensive approach, including the repeated sampling of individuals, the quantification of protozoa, and the correlation of the results with the actual health parameters of the sampled individuals. Gathering all these data is critical for the development of well-managed ecotourism and research practices that prioritize minimal impact on animal health. By monitoring and addressing the presence of these parasites in zoo animals, we can effectively mitigate the risks associated with zoonotic transmission, thereby promoting both animal welfare and public health. Specific measures include the regular cleaning of animals’ habitats, ensuring access to clean drinking water, and implementing routine deworming protocols. Therefore, it is crucial to prioritize ongoing surveillance and diagnostic measures for the detection and management of zoonotic parasites in zoo settings.

## 5. Conclusions

Our study provides valuable insights into the prevalence and genotype distribution of *Cryptosporidium* spp., *G. duodenalis*, *E. bieneusi*, and *Blastocystis* spp. infections among zoo animals in five zoos in China. Those protozoa were detected in all the zoos studied. The detection of multiple zoonotic genotypes in animals indicates the potential for zoonotic transmission to humans. These findings emphasize the importance of the regular testing of animals residing in zoo facilities in order to diagnose and prevent the spread of zoonotic parasites.

## Figures and Tables

**Figure 1 animals-14-00853-f001:**
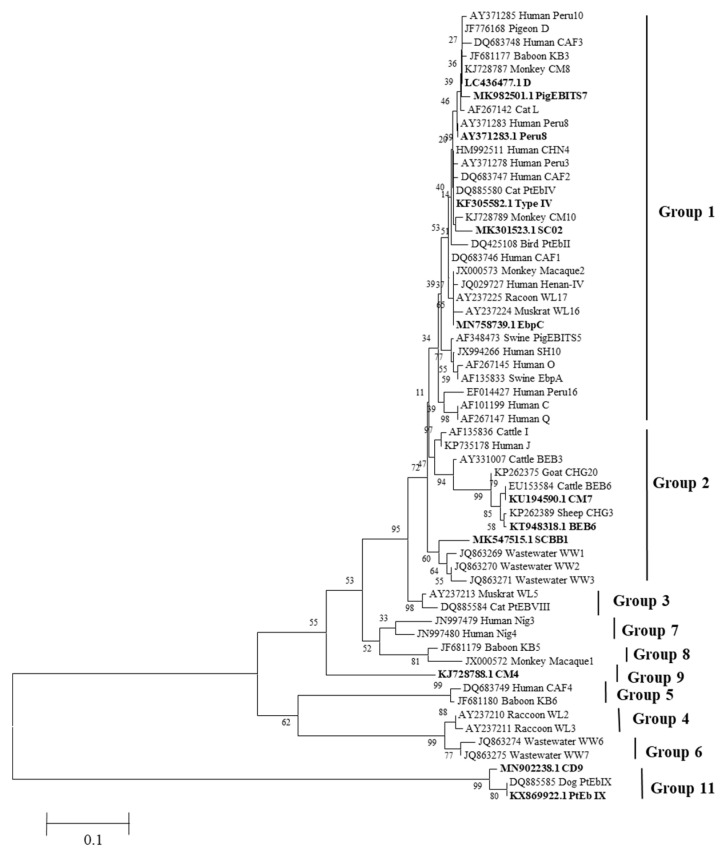
Phylogenetic relationships for *E. bieneusi* ITS sequence from the present study. Numbers on the branches are percent bootstrapping values from 1000 replicates.

**Table 1 animals-14-00853-t001:** Primers for four intestinal protozoa.

Pathogen	Gene Locus	Prime Sequence (5′-3′)	Expected Product Size (bp)	Annealing Temperature (°C)	References
*Cryptosporidium* spp.	*SSU rRNA*	SSU-F2: TTCTAGAGCTAATACATGC	~1325	55	[15]
		SSU-R2: CCCATTTCCTTCGAAACAGGA			
		SSU-F3: GGAAGGGTTGTATTTATTAGATAAAG	~840	55	
		SSU-F4: CTCATAAGGTGCTGAAGGAGTA			
*G. duodenalis*	*tpi*	AL3543: AAATIATGCCTGCTCGTCG	~605	55	[16]
		AL3546: CAAACCTTITCCGCAAACC			
		AL3544: CCCTTCATCGGIGGTAACTT	~530	55	
		AL3545: GTGGCCACCACICCCGTGCC			
	*bg*	G7: AAGCCCGACGACCTCACCCGCAGTGC	753	55	
		G759: GAGGCCGCCCTGGATCTTCGAGACGAC			
		B-F: GAA CGA ACG AGA TCG AGG TCCG	511	55	
		B-R: CTCGACGAGCTTCGTGTT			
*Blastocystis* spp.	*SSU rRNA*	RD5: ATCTGGTTGATCCTGCCAGT	~600	58	[17]
		BhRDr: GAGCTTTTTAACTGCAACAACG			
*E. bieneusi*	*ITS*	EBITS3: GATGGTCATAGGGATGAAGAGCTT	~435	57	[18]
		EBITS4: TATGCTTAAGTCCAGGGAG			
		EBITS1: AGGGATGAAGAGCTTCGGCTCTG	~392	55	
		EBITS2.4: AGTGATCCTGTATTAGGGATATT			

**Table 2 animals-14-00853-t002:** Genotype distribution of *G. duodenalis* in different zoos.

Location	Sample Size	No. of Positive Samples	Type of Assemblage		Genotype
			*tpi* (*n*)	*bg* (n)	
Guiyang	49	8	E (2), B (4)	E (5)	E (4), B (2), E + B (2)
Beijing	101	10	B (5)	E (5), B (3)	E (5), B (5)
Shijiazhuang	69	2	B (1)	E (1), B (1)	E (1), B (1)
Tangshan	66	3	B (2)	E (1)	E (1), B (2)
Xingtai	115	1	-	D (1)	D (1)
Total	400	24	E (2) B (12)	E (12), B (4), D (1)	E (11), B (10), E + B (2), D (1)

**Table 3 animals-14-00853-t003:** Distribution of *G. duodenalis* genotype in animals.

Groups of Animals	Animal Species (Common Name/Scientific Name)	Sample Size	No. of Positive Samples	Type of Assemblage
Artiodactyla	Giraffe/*Giraffa camelopardalis*	5	2	E (2)
	Milk goat/*Capra hircus* L.	3	2	E (1), E + B (1)
	Siberian ibex/*Capra sibirica*	11	2	E (2)
	Argali/*Ovis ammon*	15	1	E (1)
	Roe deer/*Capreolus pygargus*	4	2	E (2)
	Addax/*Addax nasomaculatus*	3	1	E (1)
	Yak/*Bos grunniens domesticus*	5	1	E (1)
Carnivora	Masked civet/*Paguma larvata*	3	1	D (1)
	Brown bear/Ursus arctos	6	1	E + B (1)
Primate	Ring-tailed lemur/*Lemur catta*	21	9	E (1), B (8)
	Chimpanzee/Pan troglodytes	9	2	B (2)

**Table 4 animals-14-00853-t004:** Infection rates of *Blastocystis* spp. in animals.

Groups of Animals	Animal Species (Common Name/Scientific Name)	Sample Size	No. of Positive Samples	Positive Rate (%)
Proboscidean	Asian elephant/*Pan troglodytes*	6	1	16.7
Artiodactyla	Camel/*Camelus bactrianus*	10	3	30.0
	Milk goat/*Capra hircus* L.	3	1	33.3
	Giraffe/*Giraffa camelopardalis*	5	2	40.0
	Siberian ibex/*Capra sibirica*	11	5	45.5
	Ammotragus/*Ammotragus lervia*	3	3	100.0
	Argali/*Ovis ammon*	15	6	40.0
	Roe deer/*Capreolus pygargus*	4	2	50.0
	River deer/*Hydropotes inermis*	11	9	81.8
	Addax/*Addax nasomaculatus*	3	3	100.0
	Giant eland/*Tragelaphus derbianus*	7	3	42.9
	Alpaca/*Vicugna pacos*	24	3	12.5
	Red deer/*Cervus canadensis*	18	7	38.9
	Yak/*Bos grunniens domesticus*	5	3	60.0
	Fallow deer/*Dama dama*	13	4	30.8
	Blue sheep/*Pseudois nayaur*	5	3	60.0
	Sika deer/*Cervus nippon*	18	2	11.1
	Big-eared sheep/*Capra hircus*	6	2	33.3
Perissodactyla	Zebra/*Equus quagga*	23	2	8.7
Primates	Golden monkey/ *Rhinopithecus*	5	2	40.0
	Mandrill/*Mandrillus sphinx*	9	7	77.8
	Ring-tailed lemur/ *Lemur catta*	21	11	52.4
	Gibbon/*Hylobatidae*	10	2	20.0
	Baboon/*Papio*	1	1	100.0
	Spider monkey/*Ateles*	1	1	100.0
	Black langur/ *Trachypithecus francoisi*	2	1	50.0
	Chimp/*Pan troglodytes*	9	1	11.1
	Patas monkey/*Erythrocebus patas*	3	1	33.3
	Macaque/*Macaca mulatta*	10	3	30.0
Carnivora	Ocelot/*Prionailurus bengalensis*	13	1	7.7
	Manchurian tiger /*Panthera tigris ssp. altaica*	6	1	16.7
	Gray wolf/*Canis lupus*	16	1	6.3
Avian	Cassowary/*Casuarius unappendiculatus*	1	1	100.0

**Table 5 animals-14-00853-t005:** Subtype distribution of *Blastocystis* spp. in animals.

Gene Subtypes (Population Size)	Host (Population Size)
ST1 (16)	Mandrill (7), ring-tailed lemur (4), cassowary (1), gibbon (1), spider monkey (1), chimp (1), patas monkey (1)
ST2 (6)	Ring-tailed lemur (1), baboon (1), gibbon (1), macaque (3)
ST3 (3)	Asian elephant (1), patas monkey (1), gray wolf (1)
ST4 (3)	Ring-tailed lemur (3)
ST5 (21)	Milk goat (1), Siberian ibex (1), Ammotragus (1), argali (2), river deer (3), ocelot (1), ring-tailed lemur (1), zebra (2), yak (3), fallow deer (1), blue sheep (3), red deer (1)
ST8 (3)	Camel (1), golden monkey (1), ring-tailed lemur (1)
ST10 (31)	Camel (2), giraffe (2), Siberian ibex (1), Ammotragus (1), argali (4), roe deer (1), river deer (1), addax (3), giant eland (3), fallow deer (3), alpaca (1), sika deer (2), big-eared sheep (2), red deer (5)
ST13 (2)	Golden monkey (1), black langur (1)
ST14 (12)	Siberian ibex (3), Ammotragus (1), roe deer (1), river deer (5), alpaca (2)
Unknown ST (1)	Red deer (1)

**Table 6 animals-14-00853-t006:** Occurrence of parasites among various animals in different zoos.

Locations	Sample Size	Infection Rate (%) (Positive Samples/Total Samples)				
		Total	*Cryptosporidium* spp.	*E. bieneusi*	*G. duodenalis*	*Blastocystis* spp.
Guiyang	49	55.1 (27/49)	2.0 (1/49)	12.2 (6/49)	16.3 (8/49)	24.5 (12/49)
Beijing	101	53.5 (54/101)	1.0 (1/101)	6.9 (7/101)	9.9 (10/101)	35.6 (36/101)
Shijiazhuang	69	36.2 (25/69)	0 (0/69)	11.6 (8/69)	2.9 (2/69)	20.3 (14/69)
Tangshan	66	19.8 (13/66)	0 (0/66)	6.1 (4/66)	4.5 (3/66)	10.6 (7/66)
Xingtai	115	51.3 (59/115)	0 (0/115)	25.4 (29/115)	0.9 (1/115)	25.2 (29/115)
Total	400	44.5 (178/400)	0.5 (2/400)	13.5 (54/400)	6 (24/400)	24.5 (98/400)

## Data Availability

The sequences that support the findings of this study are openly available in the GenBank database at https://www.ncbi.nlm.nih.gov/nucleotide/ (accessed on 3 June 2021).

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
