# Peer review of "Epidemiology and Molecular Characterization of Zoonotic Gastrointestinal Protozoal Infection in Zoo Animals in China"

_animals, 2024, doi:10.3390/ani14060853_

Round 1
Reviewer 1 Report
Comments and Suggestions for Authors
The manuscript needs revision. Please refer to comments given in the text of reviewed attached file of the manuscript.

Reviewer 2 Report
Comments and Suggestions for Authors
The work titled "Epidemiology and molecular characterization of zoonotic gastrointestinal protozoal infection in zoo animals in China" denotes a great effort and collected a good number of samples. It is coherent and the results and the discussion is well demonstrated and explained.
However, a geographical analysis using a map of the locations, would show with greater accuracy where exist the 4 protozoa studied. And if there's any epidemiological link.
It was also important to hypothesize what forms of transmission to humans in places where there is positivity. Since we talk about zoonotic potential, this risk assessment analysis should be done.
In the discussion/conclusions one should make a more concise and exhaustive reflection of why in these 400 samples the most found protozoas are not the ones that the literature/evidence in public health find most often and prioritize as of the highest risk. Because Cryptosporidium and Giardia are the most and in this study they are not the most prevalent.
Comments on the Quality of English Languagerevise the little minor English mistakes, please.
Reviewer 3 Report
Comments and Suggestions for Authors
The manuscript, “Epidemiology and molecular characterization of zoonotic gastrointestinal protozoal infection in zoo animals in China,” outlines the primary diseases in China's zoos. The findings revealed high intestinal parasitic protozoal infections such as Cryptosporidium spp, Giardia duodenalis, Enterocytozoon bieneusi, and Blastocystis spp. However, although the prevalence is low, Blastocystis (24.5%) is notorious, a cosmopolitan intestinal parasite inhabiting the intestinal tract of humans and numerous animals, both homeothermic and poikilothermic. I would like the authors to elaborate more on this disease's presence and its possible zoonotic effect.
On the other hand, the prevalence and genotype distribution of E. bieneusi are interesting and show phylogenetic relationships in the sequences reported. According to lines 309-312, zoonotic genotypes were identified in various animals, suggesting that zoo animals may be important sources of human E. bieneusi infections and should be closely monitored and controlled. Could the authors explain a little bit more?
In summary, this manuscript has the potential to be published in the journal Animals. I have just a few comments to be more specific on some points of the manuscript.
Author Response
We express our sincere gratitude to the reviewer for their meticulous examination of our manuscript. Upon reevaluation, we acknowledge that our initial description of the hazards and potential zoonotic risks associated with Blastocystis spp. was indeed lacking in detail. Therefore, we have made revisions to address this deficiency, which can be found on page 11, lines 333-342.
Additionally, we recognize the importance of providing comprehensive information on E. bieneusi, as highlighted by the reviewer. While this pathogen has been discussed previously on page 2, lines 82-87, we have supplemented this information with further elaboration on page 11, lines 321-323. We understand that the level of detail may not be exhaustive, but we have endeavored to strike a balance within the constraints of the paper's structure. We genuinely appreciate the reviewer's valuable comments, which have prompted us to enhance the clarity and completeness of our manuscript.

Reviewer 4 Report
Comments and Suggestions for Authors
In the context of the global spread of pathogens, the reduction of the created “barriers” isolating people and animals, new pathogens are emerging, the area with the addition of wild and domestic animals, both on farms and in laboratories and zoos, requires new approaches and research, control and changes in conditions content. .
The authors of the articles conducted a study of zoos identifying pathogens, including Blastocystis spp., E. bieneusi, Cryptosporidium sp., Giardia duodenalis.
The article contains notes:
1. Data must be summarized, including by type of animals kept.
2 It may be necessary to pay attention to the peculiarities of keeping animals, for example, cleaning and disinfecting bathing containers. Since, for example, Blastocystis spp. has not been found in either rodents or marsupials.
It should be noted that water has also been associated with the spread of E. bieneusi, through ponds, ditches and other surface waters, and several microsporidia species can be isolated from such sources, indicating that the disease may be waterborne. Thus, the genus Cryptosporidium includes 30 species, exclusively found in water or food, with high pathogenicity for humans. The group probably requires more in-depth study, taking into account the identification of species (groups of species). The identified taxon is Giardia duodenalis, also associated with additives and contact with untreated water, food, soil and contaminated fecal substrates. Infection is also possible when kept without an enclosure, like a safari park.
3. It is necessary to improve the design and presentation of the material in general, with some of the data presented in the form of graphs and diagrams:
Improvement of design based on the results of the study, provide a table:
- Distribution of pathogens by risk, distribution, transmission routes
- Provide recommendations for reducing morbidity, including taking into account the conditions of detention
Author Response
1.Thank you for your appreciation of our comments regarding the data processing. We noted your suggestion to conduct statistics based on the types of animals. Upon review, we found that Table 3 and Table 4 in our study already present statistics categorized by the types of animals. Therefore, we have not made any changes in response to this suggestion. We appreciate your understanding and attention to this matter.
2.Thank you for your valuable comments. We have incorporated the helpful suggestions into our introduction section, which can be found on page 2 line 61-74. We genuinely appreciate your acknowledgment of the value in our feedback.
3.Thank you for your valuable feedback. We regret to inform you that we are unable to provide the requested table detailing risk factors, distribution, and transmission routes. There are several reasons for this decision. Firstly, the precise risk factors for infection with these parasites remain unknown, and we lack information regarding the water availability in the zoos under study. Secondly, we have already presented statistical data on the distribution of these protozoa in Table 6. Thirdly, while the transmission routes for these protozoa primarily involve fecal-oral transmission, with some also transmitted via contaminated food and water, the specific data required for statistical analysis is unavailable.
Additionally, we appreciate your suggestion to include recommendations for reducing morbidity. Although this aspect was not initially addressed in our study, we have now incorporated relevant information into the manuscript (page 12, lines 369-373).
Once again, we sincerely appreciate your thoughtful input and constructive criticism.

Round 2
Reviewer 4 Report
Comments and Suggestions for Authors
Thank you for your feedback and detailed answers.
I propose to publish the data noted in the group with different risks of infection associated with the conditions of detention. Previously "3. It is necessary to improve the design and presentation of the material as a whole, with some of the data presented in the form of graphs and diagrams: Improved design based on the results of the study should be presented in table form: - Distribution of pathogens by risk. , distribution, transmission routes."
Point 3. I agree that additional data is necessary for analysis. However, it is possible to subdivide the studied animals into risk groups associated with the conditions of detention and/or resistance to pathogens of specific animal species. Please pay attention to this, since the entry of pathogens through food or living conditions requires preventive measures and further research into the issue, including those related to possible infection of zoo visitors. The above also emphasizes the relevance of your research. I propose to indicate the degree of zoonosis of pathogens found in the zoo, for example, according to published data, indicating groups of animals in the “risk zone”.
